# A Study on the Compaction Behavior and Parameter Sensitivity of Curing Phenolic Thermal Protection Material Strips

**DOI:** 10.3390/polym17081045

**Published:** 2025-04-12

**Authors:** Xiaodong He, Zeyu Pan, Hualian Li

**Affiliations:** College of Mechanical Engineering, Inner Mongolia University of Technology, Hohhot 010051, China; hexdong@imut.edu.cn (X.H.); 20231100018@imut.edu.cn (Z.P.)

**Keywords:** phenolic insulation material, film winding process, curing kinetics, fiber volume fraction void evolution, the Box–Behnken Design (BBD) method

## Abstract

This study investigates the curing and compaction behavior of filament–wound phenolic thermal protection materials. The optimal heating profile was determined based on curing kinetics obtained through DSC analysis. A Box–Behnken design was employed to evaluate the effects of curing temperature, pressure, and heating rate on the fiber volume fraction. Microscopic analysis revealed the evolution and gradient distribution of voids during the curing process. A regression model was established, and sensitivity analysis revealed that curing pressure had the greatest influence, followed by heating rate and temperature. These findings offer insights into void control mechanisms and process optimization strategies for high-performance phenolic composites.

## 1. Introduction

Phenolic thermal protection materials are among the most established thermal protection systems due to their excellent thermal insulation, mechanical strength, and ablation resistance. These materials are widely employed in aerospace applications for thermal protection [1,2,3,4]. Since the development of the first filament winding machine in the United States in 1947 for manufacturing solid rocket motor casings, filament winding has evolved into a key manufacturing technique for phenolic thermal protection materials. Today, it is widely used in the fabrication of solid rocket motor nozzles and cylindrical components [5]. Filament winding processes are classified into two types based on the material form: fiber winding and prepreg tape winding. In fiber winding, continuous fibers are wound onto a predesigned mold surface under controlled winding angles and tension. Based on the resin state during winding, fiber winding can be further classified into wet, semi-dry, and dry methods [6]. The curing and compaction of wound preforms represent a critical stage in the tape-winding process. By analyzing the curing process, complex behavior can be simplified into determining the required pressure and temperature for product formation. The temperature field during the curing of resin matrix composites is essentially an anisotropic heat conduction problem with a nonlinear internal heat source. Loos A.C. et al. [7] neglected convective heat transfer and developed a heat conduction–curing model based on Fourier’s law for thermosetting resin composites, offering a method to determine appropriate curing cycles. Lee et al. [8,9] analyzed the curing degree and viscosity of epoxy resin using differential scanning calorimetry (DSC) and rheological techniques and derived a viscosity model that aligned well with experimental results. S. Yi et al. [10] solved the heat conduction equation using the finite element method, revealing that laminate thickness significantly influences thermal conductivity and enabling prediction of temperature and curing degree distributions. Wang Junmin et al. [11] proposed an optimization approach to achieve uniform curing during compression molding. Finite element simulations demonstrated that this approach enhances the uniformity of the curing temperature field.

The pressure field during the curing process primarily influences resin flow and fiber compaction. Due to the absence of in situ monitoring devices, the curing process is typically analyzed based on theoretical assumptions and numerical simulations. Loos and Springer proposed a layer-by-layer compaction model, assuming that resin matrix composites behave as incompressible porous media. In this model, resin flow along the fiber direction is described by a pipe flow model, while flow perpendicular to the fibers follows Darcy’s law. The resin is assumed to carry all external loads [12]. Building on this, Gutowski et al. [13,14] developed a progressive compaction model, also known as the “squeezing sponge model”, in which both the fiber network and resin share the applied load. At the early stage of curing, the resin carries the entire load. As compaction progresses and resin flow increases, the fiber network gradually begins to support the load. By the end of the curing process, the resin no longer carries any load, and the fibers bear all applied forces, making this model more representative of real processing conditions.

Curing kinetics analysis primarily relies on thermal analysis techniques, including differential scanning calorimetry (DSC), Fourier-transform infrared (FTIR) spectroscopy, and thermogravimetric analysis (TGA). Among these techniques, DSC is most widely used to monitor the entire resin curing process. Dynamic and isothermal DSC are commonly used to study curing mechanisms, where experimental data are fitted to empirical equations to establish curing kinetics models [15,16]. Numerous studies have examined the curing behavior of filament–wound structures. Yang Hui [17], based on Springer’s heat conduction model, performed numerical simulations to analyze the effect of winding speed on temperature and degree of cure distribution, showing that excessive winding speed results in incomplete curing. Ren Mingfa et al. [18] investigated fiber compaction during the filament winding curing process through numerical simulations under varying winding tensions. The results indicated that maintaining uniform tension during winding promotes even resin distribution. A critical prerequisite for the application of phenolic thermal protection materials lies in their forming and manufacturing processes.

A high fiber volume fraction and low porosity are essential for achieving superior performance in phenolic thermal protection materials. During curing, phenolic resins undergo condensation reactions that release volatile by-products, such as water vapor. If these gases are not promptly removed, micropores can form within the material, significantly compromising its mechanical and thermal protective properties [15,19]. To reduce porosity, vacuum bag molding or compression molding is commonly used in engineering practice for compaction and degassing [20]. Studies have shown that inadequate degassing prior to pressurization can lead to bubble entrapment after resin gelation, ultimately resulting in pore structures in the final product that are difficult to eliminate [19,21]. This issue is particularly severe in thick-walled structures, where residual porosity becomes a key factor limiting product consistency and reliability [22]. However, existing studies primarily focus on pore characterization at the later stages of curing, providing limited insight into the continuous evolution of pore morphology and distribution throughout the entire curing process. In contrast, in other thermosetting composite systems, such as epoxy, in situ imaging studies using techniques like X-ray computed tomography (CT) have been extensively conducted, whereas such investigations remain exceedingly rare for phenolic systems. Therefore, it is imperative to investigate the mechanisms of pore evolution during the curing of phenolic resins to improve process control and ensure final product quality.

In tape-winding processes, the preform experiences structural changes under the combined influence of winding pressure, tension, and temperature. During curing, the preform is compacted under elevated temperature and pressure, and the selection and control of process parameters directly determine the final product’s quality and reliability [12,23]. Shi Y.y et al. [24] investigated the tape-winding process and identified winding pressure, tension, and temperature as the key parameters influencing product quality. Their experimental results demonstrated that product quality is affected by the coupling effects of these parameters. Although composite resin systems are diverse, research on curing kinetics remains limited, particularly regarding the curing and compaction behavior of phenolic resins. This study investigates the curing and compaction behavior in tape-winding using mechanistic modeling and experimental validation. The influence of process parameters on product performance is analyzed, and the sensitivity of product properties to curing conditions is quantitatively assessed. This approach provides a reference for selecting and controlling process parameters in manufacturing and optimizing the forming process.

## 2. Materials and Methods

### 2.1. Experimental Materials and Equipment

The phenolic prepreg tape used in this study was fabricated using a 2130 phenolic resin system (Beijing, China), specifically formulated for thermal protection applications. As shown in Figure 1, the phenolic prepreg tape adopts a plain-woven architecture, as observed in the optical micrograph (Figure 1a). The schematic diagram (Figure 1b) illustrates the multilayer interlaced structure through the thickness. The tape has a width of 82 mm and an average single-layer thickness of 0.22 mm. Auxiliary materials used in the curing experiments included Teflon release film (Yilifeng Technology Co., Ltd., Dongguan, China), LG230 sealing tape (Shanghai, China), VB200 vacuum bag film (Easy Composites Ltd., Stoke-on-Trent, UK), and a polyester fiber breather mat (Suzhou Qicai Stone Co., Ltd., Suzhou, China). A silicone rubber heating pad (200 × 300 mm, Manying Electrical Co., Ltd., Shenzhen, China) was employed for localized heating.

The key experimental equipment includes a Q20 differential scanning calorimeter (TA Instruments, New Castle, DE, USA), with aluminum crucibles supplied by Shanghai Hesheng. The curing process was conducted in a custom-built autoclave (Φ0.5 m × 0.8 m, Suzhou Ruixiang Electronics Co., Ltd., Suzhou, China), equipped with a digital temperature controller (XMTA-7000, Jingle Home, Wenzhou, China). Pressure and vacuum were regulated using a rotary vane vacuum pump (XZ-1, Yancheng, China), an air pump (Y-128, Ningbo, China), and a pressure gauge system (BR3000, Huashi Electrical Instrument Co., Ltd., Wuxi, China). Additional equipment includes an MB120 cutting machine, a 0.001 g precision electronic balance (Coolbay Instruments Co., Ltd., Shenzhen, China), an optical microscope (JX13C, Guiyang Xintian Optical Instrument Co., Ltd., Guiyang, China), and a muffle furnace (DYXL-600, Hebi Electronics Research Institute, Hebi, China). The physical state parameters of the protective materials are listed in Table 1.

### 2.2. Experiment

#### 2.2.1. DSC Experiment

Approximately 5 mg of phenolic resin liquid was placed in a pierced aluminum crucible and left to stand until its weight stabilized to ensure complete solvent evaporation. Thermal scans were conducted using a TA Instruments Q20 DSC under a nitrogen flow of 20 mL/min, with a temperature ramp from 20 °C to 250 °C at heating rates of 5, 10, 15, and 20 °C/min. The curing enthalpy and characteristic temperatures of the resin were recorded to determine the optimal curing parameters.

#### 2.2.2. Test Piece Preparation and Curing Experiment

As shown in Figure 2, the curing platform integrates pressure regulation, air supply, vacuum evacuation, and thermal control systems. Laminates were laid up and sealed in vacuum bags with release films, breather layers, and adhesive felt, then cured in the autoclave. Pressure was regulated using an air pump and monitored with a pressure gauge, while vacuum was maintained throughout the curing process to prevent volatile retention. After curing, the specimens were trimmed to the required dimensions for subsequent testing.

### 2.3. Characterization

#### 2.3.1. Experimental Design

A Box–Behnken design (BBD) was employed to evaluate the effects of curing temperature (150–180 °C), heating rate (1–6 °C/min), and pressure (0.1–0.6 MPa) on the fiber volume fraction. The factor levels for the experimental design are listed in Table 2.

For void analysis, a single-factor experimental approach was employed to observe void distribution under various curing conditions. The experimental factor levels are listed in Table 3.

#### 2.3.2. Fiber Volume Fraction Measurement

The fiber volume fraction was measured using a pyrolysis method, in which the specimens were heated in a muffle furnace from room temperature to 800 °C at a rate of 10 °C/min and held for 1 h. The thermal degradation process—including the initial state, heating phase, and post-degradation residue—is illustrated in Figure 3.

#### 2.3.3. Void Distribution Observation

The void structure and distribution were examined using optical microscopy on cross-sectional specimens of the laminate. Laminates cured under different processing conditions were used to investigate the effects of heating rate, curing temperature, and pressure on void morphology.

This section provides the experimental foundation for the modeling and analysis presented in Section 3, enabling the evaluation of how individual curing parameters influence compaction behavior and microstructural integrity.

## 3. Results and Discussion

### 3.1. Thermal Analysis Test Results

To investigate the thermal behavior of phenolic resin and develop a multi-stage curing process suitable for filament-wound composites, differential scanning calorimetry (DSC) analysis was performed. A precise understanding of the resin’s curing kinetics is essential for coordinating its flow behavior, viscosity evolution, and the compaction of fiber laminates. Figure 4 shows the DSC heat flow curves obtained at different heating rates (5, 10, 15, and 20 °C/min). The results indicate that the phenolic resin system exhibits a single exothermic peak across all heating rates. With increasing heating rate, the exothermic peak shifts upward from 158.7 °C to 178.3 °C. As shown in Table 4, the initial temperature ($T_{i}$), peak temperature ($T_{p}$), and final temperature ($T_{f}$) all increase with heating rate, whereas the total exothermic enthalpy (∆H) decreases from 15.90 J/mg to 12.36 J/mg. This phenomenon suggests that at higher heating rates, the rapid temperature rise induces thermal lag, hindering complete resin curing during the initial stage.

This thermal lag behavior has been widely reported in previous studies. Dusi et al. [8] and Guo et al. [15] reported that higher heating rates lead to more delayed and sharper exothermic peaks in thermosetting systems such as phenolic and epoxy resins. This is primarily because the external heat input rate exceeds the intrinsic reaction rate, resulting in thermal lag. Park et al. [25] further noted that the exothermic peak of resol-type phenolic resin can shift by over 20 °C as the heating rate increases, and the total heat release decreases due to kinetic limitations. In addition, the degree of cure at the exothermic peak ($\alpha_{p}$) remains between 0.46 and 0.52, indicating that the system does not achieve complete curing at the peak temperature. The observed reduction in Δ***H*** and the narrowing of the reaction window (Δ***T***) further highlight the importance of rational heating program design to avoid premature gelation or incomplete curing.

Based on the above results, a two-stage curing process was proposed to ensure adequate resin impregnation while enabling complete cross-linking reactions. The recommended curing schedule consists of two stages: 130 °C for 30 min to promote resin flow and impregnation, followed by 165 °C for 2 h to complete the cross-linking reaction. The corresponding temperature–time profile is illustrated in Figure 5. Compared with conventional empirical approaches, the DSC-based curing path offers a scientific basis for controlling the curing process and is particularly suitable for optimizing the molding of high-performance phenolic resin systems in aerospace thermal protection applications.

### 3.2. Analysis of the Impact of Curing Process Parameter Coupling on Fiber Volume Fraction

To investigate the influence of key curing process parameters on the fiber volume fraction of phenolic thermal protection laminates, an experimental scheme based on the Box–Behnken Design (BBD) was developed. The selected process variables include curing temperature (*T*), heating rate (*Tv*), and curing pressure (*P*). According to the standard method for measuring fiber volume fraction in phenolic laminates, specimens prepared under different parameter combinations were tested, and the results are summarized in Table 5.

The experimental data were modeled and analyzed via regression using Design-Expert software (Version 13, Stat-Ease Inc., Minneapolis, MN, USA). The fitting performance of various regression models is summarized in Table 6.

As shown in Table 6, among all evaluated models, the quadratic regression model exhibited the best statistical performance, with both the adjusted R^2^ and predicted R^2^ exceeding 0.93. Furthermore, the lack-of-fit *p*-value was statistically insignificant (*p* > 0.05), indicating that the model fits the system’s response behavior well. Accordingly, a quadratic regression model was established to describe the relationship between fiber volume fraction (V) and curing temperature (*T*), heating rate ($T_{v}$), and pressure (*P*), as shown in Equation (1).(1)$$V_{f}=-6.393+0.471T+5.8T_{v}+73.07P-0.012T\times T_{v}\;-0.245T\times P+0.176T_{v}\times P-7.178\times 10^{-4}T^{2}-0.474{T_{v}}^{2}-29.344P^{2}$$

To validate the reliability of the experimental data and the accuracy of the fitted model, additional analysis was performed using Design-Expert software. This included residual analysis, analysis of variance (ANOVA), and comparison between predicted and actual values. The normal probability plot of residuals and the comparison between predicted and actual fiber volume fraction values are presented in Figure 6.

To further assess the reliability of the model, residual analysis and a predicted-versus-actual value comparison were performed, as shown in Figure 6. As shown in Figure 6a, the residuals are approximately aligned along a straight line, indicating that they follow a normal distribution without significant outliers. In Figure 6b, most predicted values are closely distributed around the 1:1 reference line, indicating good agreement between predicted and experimental results. This confirms both the high predictive accuracy of the model and the reliability of the experimental data.

Subsequently, a one-way analysis of variance (ANOVA) was conducted to determine the significance of each process parameter. The model yielded an F-value of 48.95, and the *p*-values for curing temperature, heating rate, and pressure were all below 0.05, indicating that all three parameters have a statistically significant effect on the fiber volume fraction.

Figure 7, Figure 8 and Figure 9 illustrate the interactions among the three key process parameters. As shown in Figure 7, at low curing temperatures and low heating rates, high resin viscosity and slow curing progression hinder resin flow and fiber impregnation, resulting in a reduced fiber volume fraction. As the curing temperature increases, resin viscosity decreases, promoting improved fiber impregnation and matrix compaction. However, both excessively high and low heating rates can reduce the fiber volume fraction. This phenomenon is attributed to premature surface gelation at high heating rates and uneven curing at low heating rates.
Figure 7Interaction effect of curing temperature and heating rate on the fiber volume fraction of phenolic thermal protection laminates.
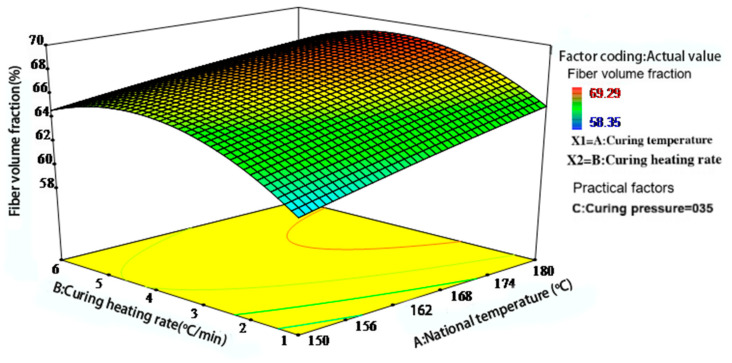



Figure 8 highlights the critical influence of curing pressure. At moderate curing temperatures, appropriately increasing the pressure effectively reduces the flow resistance caused by high resin viscosity, eliminates voids, and improves compaction. When both curing temperature and pressure are relatively high (e.g., 165–180 °C and 0.4–0.6 MPa), a higher fiber volume fraction can be achieved.
Figure 8Interaction effect of curing temperature and pressure on the fiber volume fraction of phenolic thermal protection laminates.
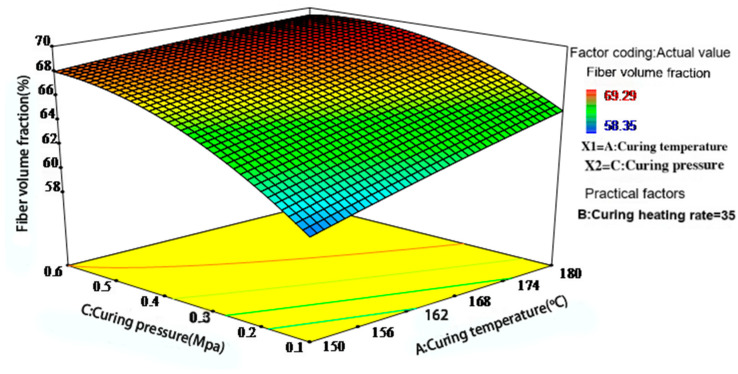



Figure 9 further illustrates the combined effect of heating rate and pressure. Under low-heating-rate and low-pressure conditions, the curing process proceeds slowly. Partial resin gelation may occur during the flow stage, hindering subsequent compaction and resulting in a lower fiber volume fraction. As pressure increases, resin outflow improves, thereby enhancing compaction. However, if the heating rate is too high, surface resin may cure prematurely, restricting internal resin flow and resulting in a pronounced curing gradient that ultimately lowers the fiber volume fraction.
Figure 9Interaction effect of heating rate and pressure on the fiber volume fraction of phenolic thermal protection laminates.
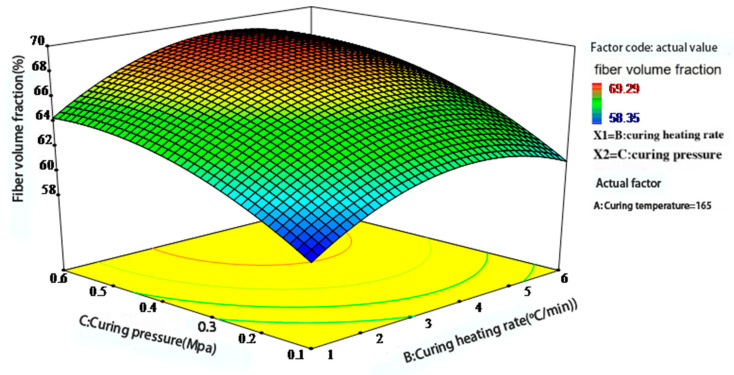



Based on the graphical results, optimal forming performance is achieved when the heating rate is maintained between 2.5 and 4.5 °C/min and the pressure is maintained between 0.4 and 0.6 MPa. Overall, the optimization of the curing process requires a coordinated balance among temperature, heating rate, and pressure, with pressure exerting the most significant influence on the fiber volume fraction. This conclusion aligns with the compaction mechanism of laminated composites proposed by Gutowski et al. [14,21].

### 3.3. Analysis of the Effect of Curing Process Parameters on Void Distribution

The void distribution and morphology in the cured phenolic resin laminates were characterized using optical microscopy. As shown in Figure 10, Figure 11 and Figure 12, voids appear as black cavities in the optical micrographs. Two predominant types of voids were observed in all specimens: small circular voids and elongated elliptical voids aligned along the fiber direction. These morphological characteristics suggest distinct formation mechanisms, such as gas entrapment and matrix shrinkage. Representative voids are highlighted in the figures using red arrows and yellow circles.

#### 3.3.1. Effect of Curing Temperature

Figure 10 shows the void distribution in phenolic laminates cured at three different temperatures: 150 °C, 165 °C, and 180 °C. The overall void morphology remains consistent, with voids primarily concentrated in the middle and upper regions along the laminate thickness direction. This gradient distribution is attributed to internal temperature and degree-of-cure gradients that develop during the curing process, particularly under non-autoclave or atmospheric pressure conditions.
Figure 10Optical micrographs showing void distribution in phenolic laminates cured at different temperatures.
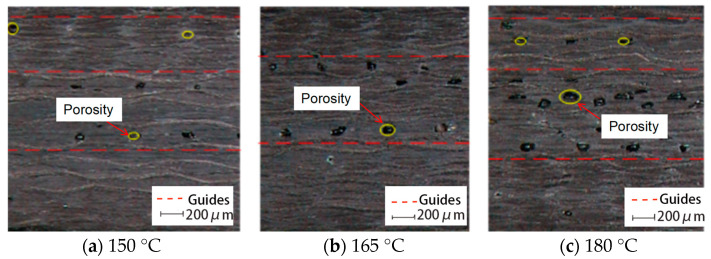



At lower curing temperatures (e.g., 150 °C), the resin remains in a low-viscosity state for a longer duration, allowing sufficient time for the release of reaction-generated gases and residual volatiles. As a result, fewer and smaller voids are observed. However, as the curing temperature increases, the region near the heat source tends to cure earlier, forming a superficially cured resin layer that acts as a barrier to internal gas release. This premature skin layer restricts gas diffusion, leading to bubble entrapment within the laminate and the formation of void-rich zones in the central and upper regions. Therefore, as shown in Figure 10c, higher curing temperatures result in a greater number of voids with larger sizes.

This phenomenon is consistent with the findings of Pupin et al. [19], who reported that in RTM epoxy systems, premature surface curing prior to gas release resulted in more severe void entrapment. Similarly, Mehdikhani et al. [22] observed that voids in thermoset composites tend to accumulate in the mid-thickness region, which is attributed to the mismatch between heat conduction and reaction rates during curing, leading to internal temperature and cure gradients.

#### 3.3.2. Effect of Heating Rate

Figure 11 illustrates the void distribution characteristics of phenolic laminates cured at different heating rates (1 °C/min, 3.5 °C/min, and 6 °C/min). At lower heating rates, as shown in Figure 11a, the resin heats gradually, allowing more time for gas release and resin flow, resulting in fewer and more uniformly distributed voids. In contrast, as the heating rate increases, the number of voids increases significantly, as observed in Figure 11b,c.
Figure 11Optical micrographs showing void distribution in phenolic laminates cured at different heating rates.
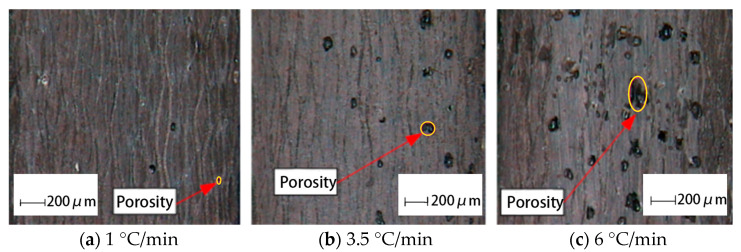



This phenomenon can be attributed to the temperature gradient-induced lag effect. At high heating rates, the resin near the mold or heat source rapidly reaches the gel point, forming a cured surface layer that encapsulates the still-uncured interior resin. This encapsulation hinders the escape of internal gases, which, under pressure, leads to the formation of elliptical or irregular voids.

This void formation mechanism aligns closely with the findings of Agius et al. [21], who reported that during rapid heating in non-autoclave composite curing, premature surface curing results in gas entrapment and the formation of numerous internal voids.

#### 3.3.3. Effect of Curing Pressure

To further investigate the role of curing pressure in void formation, optical microscopy was performed on laminates cured under varying pressure conditions (0 MPa, 0.1 MPa, 0.4 MPa, and 0.6 MPa), as shown in Figure 12.

As shown in Figure 12a,b, in the absence of external pressure or under low-pressure conditions (0.1 MPa), a large number of voids with relatively large sizes are observed in the laminates. This is primarily attributed to insufficient resin compaction, which impedes the effective evacuation of gases and volatiles before gelation.

As the curing pressure increases to 0.4 MPa and 0.6 MPa (Figure 12c,d), both the void content and size are significantly reduced. This indicates that applying appropriate pressure enhances resin flow, facilitates gas evacuation, and improves composite consolidation. However, the improvement becomes less pronounced when increasing the pressure from 0.4 MPa to 0.6 MPa, indicating a clear marginal effect. This phenomenon is attributed to the fiber laminate structure approaching its compaction limit, beyond which additional pressure has minimal effect on void elimination. Additionally, a gradient void distribution along the thickness direction remains evident, with voids primarily concentrated in the mid-thickness region. This trend corresponds to the characteristic non-uniform pressure distribution within the resin during curing. Similar findings were reported by Gutowski et al. [14], who noted that pressure gradients during curing are a major factor contributing to void accumulation in the central regions of laminated composites.
Figure 12Optical micrographs showing void distribution in phenolic laminates cured under different pressures.
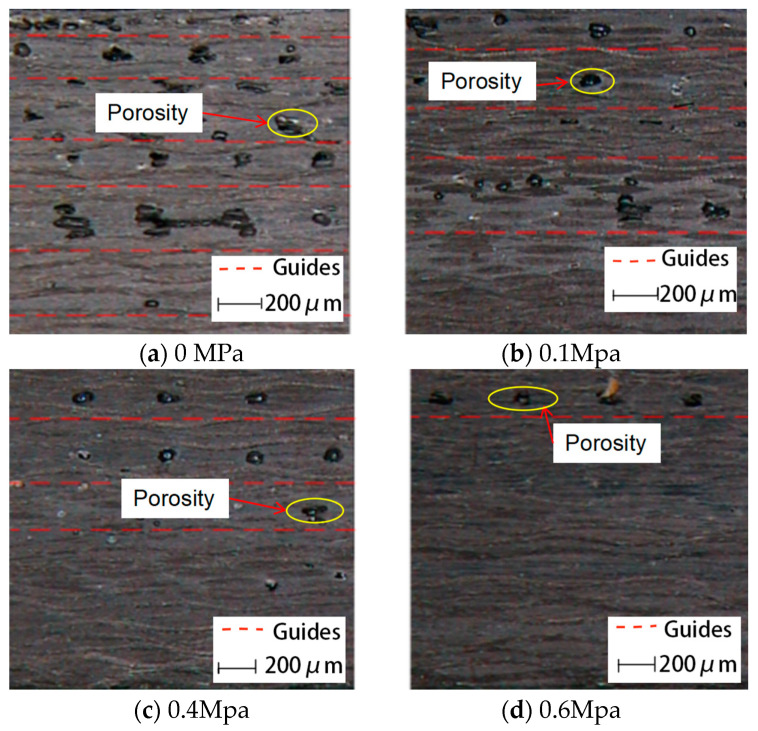



These experimental observations are consistent with classical void nucleation theories in thermosetting composites. Pupin et al. [19] reported that a substantial amount of volatile gases is released during the polycondensation reaction of phenolic resin. If these gases cannot escape in a timely manner under high-viscosity or low-pressure conditions, they may nucleate within the matrix and form closed voids. The spatial distribution and morphology of voids are collectively governed by local pressure fields, thermal gradients, and resin flow resistance.

Therefore, optimizing the coupled effects of curing temperature, heating rate, and pressure is essential for achieving a denser composite structure and meeting the stringent performance requirements of high-reliability applications, such as those in aerospace.

### 3.4. Sensitivity Analysis of Curing Process Parameters

Based on the established regression model, a sensitivity analysis was performed by calculating the partial derivatives of the fiber volume fraction with respect to the process parameters, namely curing temperature, heating rate, and pressure. The corresponding expressions are provided in Equation (2), and the calculated sensitivity results are summarized in Table 7.(2)$$\left\{\begin{array}{c} S_{V_{f}}^{T}=\frac{\partial V_{f}(\bar{T},\bar{Tv},P)}{\partial T}\\ S_{V_{f}}^{Tv}=\frac{\partial V_{f}(\bar{T},\bar{Tv},P)}{\partial Tv}\\ S_{V_{f}}^{P}=\frac{\partial V_{f}(\bar{T},\bar{Tv},P)}{\partial P} \end{array}\right.$$

$S_{V_{f}}^{i}$: sensitivity of fiber volume fraction to individual curing process parameters. $\bar{T}$, $\bar{Tv}$, and $\bar{P}$: median values of curing temperature, heating rate, and pressure within their respective parameter ranges.

As shown in Table 7, the fiber volume fraction is most sensitive to curing pressure, followed by heating rate, while curing temperature exhibits the lowest sensitivity. This ranking suggests that pressure plays a dominant role in resin compaction and void elimination during curing, making it the most critical parameter influencing material densification and forming quality.

These findings are consistent with the theoretical and experimental studies of Gutowski et al. [14,22], who emphasized that applying appropriate pressure facilitates fiber bed compaction and promotes the removal of entrapped gases. The effect of heating rate on fiber volume fraction is of moderate significance, which can be attributed to its dual influence on resin flowability and curing kinetics. In contrast, the relatively low sensitivity to curing temperature suggests that temperature variations have limited impact on compaction behavior and resin viscosity, whereas the evolution of the dynamic thermal field plays a more critical role.

### 3.5. Limitations and Future Perspectives

Although this study systematically investigated the curing behavior and compaction mechanisms of filament–wound phenolic thermal protection composites, several limitations remain and warrant further investigation in future research.

First, the experiments were conducted using flat laminate structures with constant resin formulation and fixed winding tension. However, in practical engineering applications, complex geometries—such as curved components or structures with variable thickness—often exhibit non-uniform temperature and pressure distributions during curing. Therefore, the applicability of the proposed model to non-uniform geometries requires further validation and refinement.

Second, the regression model used to predict fiber volume fraction is based on a quadratic fit of discrete experimental data points. While the model demonstrates high accuracy within the tested parameter range, it does not account for higher-order nonlinear effects, batch-to-batch material variability, or microscale phenomena such as fiber bridging and resin accumulation. These factors may significantly affect compaction behavior and void evolution during actual processing.

Third, void analysis was performed using two-dimensional optical microscopy, which, although effective for identifying void distribution and morphology, lacks three-dimensional resolution. In contrast, non-destructive 3D imaging techniques such as X-ray micro-computed tomography (μCT) can offer more comprehensive information on void volume fraction, connectivity, and spatial distribution. Future research could integrate μCT analysis with thermo-flow-mechanical coupling simulations to better elucidate the internal evolution mechanisms of voids.

Lastly, this study focused on three primary process parameters: curing temperature, heating rate, and pressure. However, other factors—such as resin volatility, ambient humidity, mold thermal conductivity, and the use of degassing or vacuum assistance—may also play significant roles in practical manufacturing scenarios. Moreover, future studies could explore the integration of in situ process monitoring techniques (e.g., fiber Bragg grating sensors, dielectric analysis) to enable real-time feedback and adaptive control during curing.

In summary, future research can be directed toward the following key areas: (1) applying the model to the forming processes of complex geometries—such as curved and variable-thickness components—to evaluate its applicability under practical engineering conditions; (2) optimizing curing paths by employing multiphysics simulations that couple thermal, flow, and mechanical fields; and (3) integrating macro-scale process modeling with microstructural evolution and property characterization to improve the engineering relevance of the findings. This integration will support quality control and facilitate the broader implementation of high-reliability aerospace composite manufacturing.

## 4. Conclusions

This study systematically investigated the curing and compaction behavior of filament-wound phenolic thermal insulation strips, with a focus on the relationships among process parameters, microstructural evolution, and product quality. Based on curing kinetics derived from DSC analysis, a stepwise heating profile was designed to align with the thermal reactivity of the phenolic resin. The optimized curing cycle was experimentally validated using a custom-designed, vacuum-assisted pressure curing platform.

A regression model based on the Box–Behnken design was established to quantify the effects of curing temperature, heating rate, and pressure on fiber volume fraction. The results showed that all three parameters significantly influenced compaction quality, with curing pressure being the most dominant factor. A novel sensitivity analysis, based on partial derivatives of the regression model, was used to rank the relative influence of each parameter, providing a practical basis for process optimization.

Furthermore, microstructural analysis revealed a gradient void distribution along the laminate thickness, attributed to the combined effects of temperature gradients, resin viscosity variation, and volatile retention. A void evolution mechanism was proposed, consistent with existing studies on volatile-induced porosity formation. The results confirmed that a moderate heating rate (2.5–4.5 °C/min) and pressure (0.4–0.6 MPa) effectively suppress void formation and enhance fiber consolidation.

Overall, this study provides both a theoretical framework and practical engineering strategies to improve the manufacturing quality of phenolic thermal protection materials. The proposed methods and insights can guide future optimization of prepreg winding and curing processes, particularly for thick-walled or geometrically complex components in aerospace thermal protection systems.

## Figures and Tables

**Figure 1 polymers-17-01045-f001:**
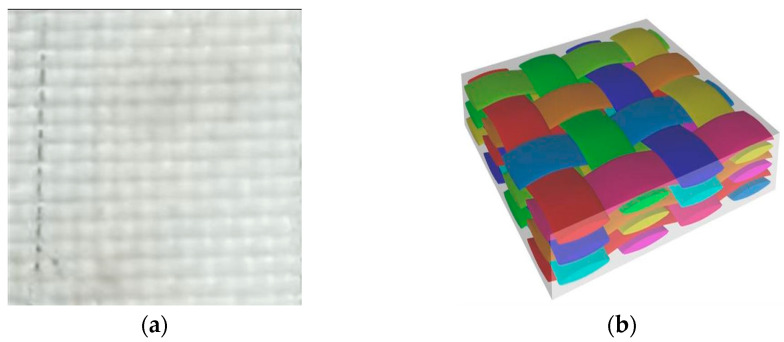
Weaving structure of phenolic prepreg tape. (**a**) Optical micrograph of the woven phenolic prepreg. (**b**) Schematic illustration of the multilayer interlaced architecture.

**Figure 2 polymers-17-01045-f002:**
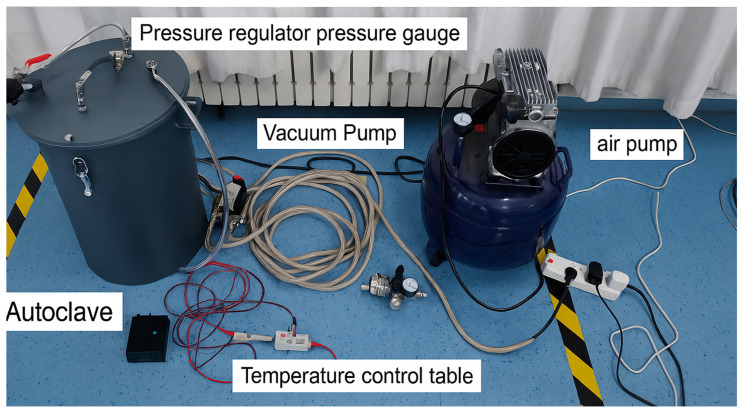
Curing equipment setup for phenolic thermal protection materials. The system includes an autoclave, vacuum pump, air pump, temperature control unit, and associated piping and gauges.

**Figure 3 polymers-17-01045-f003:**
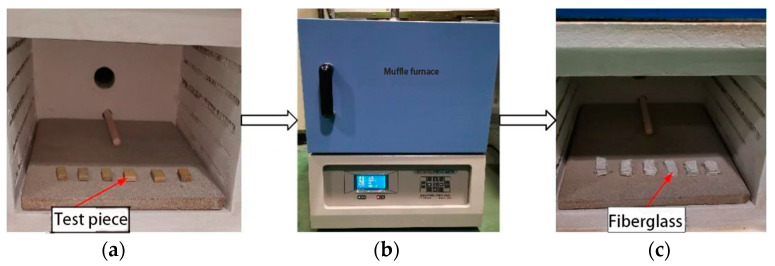
Thermal degradation procedure for phenolic thermal protection material specimens. (**a**) Specimens prior to heating. (**b**) Muffle furnace during thermal processing. (**c**) Residual glass fibers after degradation.

**Figure 4 polymers-17-01045-f004:**
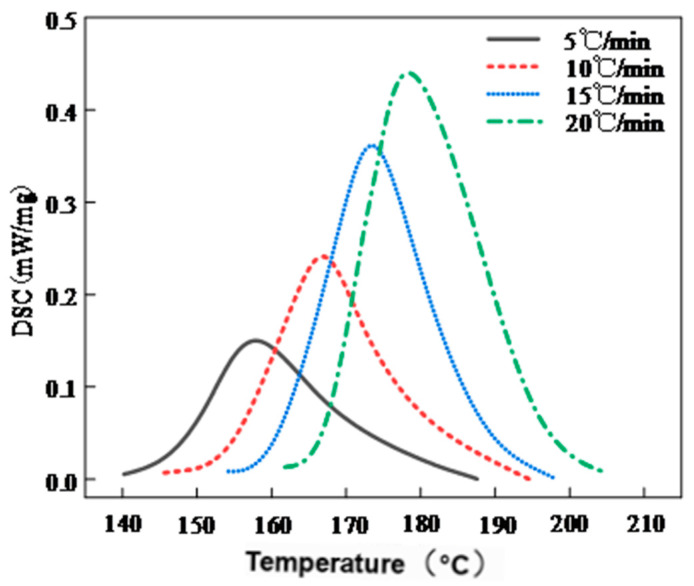
DSC heat flow curves of phenolic resin at different heating rates.

**Figure 5 polymers-17-01045-f005:**
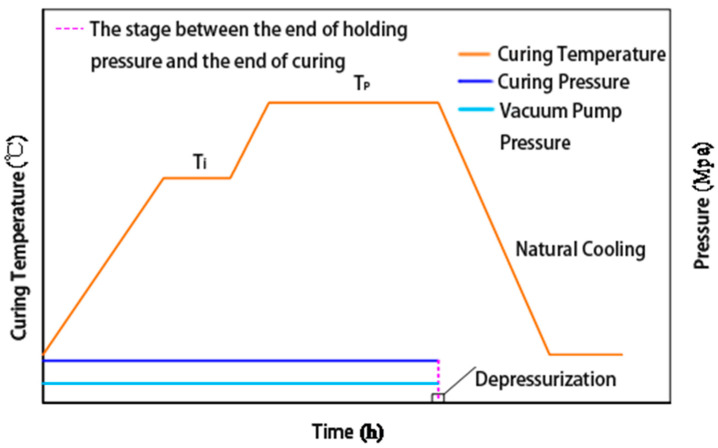
Temperature–time profile for the proposed laminate curing process.

**Figure 6 polymers-17-01045-f006:**
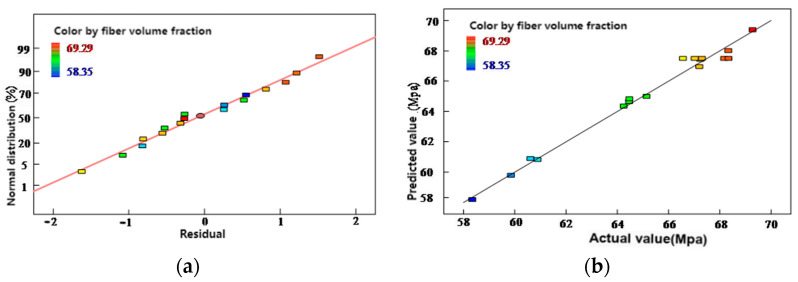
(**a**) Normal probability plot of residuals for fiber volume fraction results; (**b**) Comparison between predicted and actual values of fiber volume fraction.

**Table 1 polymers-17-01045-t001:** Physical state parameters of phenolic thermal protection materials.

Parameter	Density (kg/m³)	Specific Heat (J/(kg·K))	Thermal Conductivity (W/(m·K))
Value	1578	862	2.14

**Table 2 polymers-17-01045-t002:** Factor levels for fiber volume fraction testing of phenolic thermal protection material laminates.

Level	Factor 1: Curing Temperature (°C)	Factor 2: Curing Heating Rate (°C/min)	Factor 3: Curing Pressure/Mpa
−1	150	1	0.1
0	165	3.5	0.35
1	180	6	0.6

**Table 3 polymers-17-01045-t003:** Process parameters and factor levels for single-factor void analysis experiments.

Factor	Level			
Curing Temperature/°C	150	165	180	-
Curing Heating Rate/(°C/min)	1	3.5	6	-
Curing Pressure/Mpa	0	0.1	0.4	0.6

**Table 4 polymers-17-01045-t004:** Peak temperatures, exothermic enthalpy, and degree of cure at the peak of phenolic resin under different heating rates.

β(°C·min^−1^)	$T_{i}$*/*°C	$T_{p}$*/*°C	$T_{f}$*/*°C	∆T*/*°C	$\alpha_{p}$	∆H/(J/mg)
5	146.14	158.72	180.71	34.57	0.46	15.90
10	155.05	167.38	184.25	29.20	0.51	15.56
15	160.22	173.97	188.49	28.27	0.47	13.16
20	167.29	178.30	195.12	27.83	0.52	12.36

**Table 5 polymers-17-01045-t005:** Fiber volume fraction of phenolic thermal protection laminates under different curing parameter combinations.

Numbering	Factor 1 *T*: Curing Temperature/°C	Factor 2 *Tv*: Curing Heating Rate /(°C/min)	Factor 3 *P*: Curing Pressure /Mpa	Response: Fiber Volume Fraction/%
1	−1	1	0	64.47
2	−1	−1	0	60.62
3	0	1	−1	60.91
4	0	0	0	67.31
5	1	−1	0	65.15
6	0	0	0	68.18
7	0	−1	−1	58.35
8	0	0	0	67.03
9	−1	0	1	68.34
10	1	0	1	69.29
11	0	1	1	67.25
12	1	0	−1	64.48
13	−1	0	−1	59.86
14	0	−1	1	64.26
15	0	0	0	66.57
16	1	1	0	67.21
17	0	0	0	68.35

**Table 6 polymers-17-01045-t006:** Fitting performance of regression models for fiber volume fraction.

Model Type	Sequence *p*-Value	Lack-of-Fit *p*-Value	Adjusted R^2^	Predicted R^2^	Evaluation Result
Linear Model	0.0024	0.0159	0.5796	0.4432	-
2FI Model	0.8628	0.0092	0.4910	0.0229	-
Quadratic Model	<0.0001	0.8279	0.9643	0.9346	Optimal
Cubic Model	0.8279	-	0.9598	-	-

**Table 7 polymers-17-01045-t007:** Sensitivity of tape-winding curing process parameters.

Process Parameter	Sensitivity
Curing Temperature	0.344
Curing Heating Rate	0.573
Curing Pressure	12.774

## Data Availability

The original contributions presented in this study are included in the article. Further inquiries can be directed to the corresponding author.

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
