# Peer review of "A Study on the Compaction Behavior and Parameter Sensitivity of Curing Phenolic Thermal Protection Material Strips"

_polymers, 2025, doi:10.3390/polym17081045_

Round 1
Reviewer 1 Report
Comments and Suggestions for Authors
This study is on the curing process of filament-wound phenolic resin, which is widely used in aerospace due to its excellent insulation, mechanical properties, and ablation resistance. The study analyzes the curing kinetics to optimize the prepreg curing process and establishes a curing platform. The study provides a strategy for improving curing compaction and product quality. This article is relevant for publication, but some issues need further clarification before publication. Please answer the following questions:
- How do the observed variations in resin viscosity during the curing process influence the final mechanical properties and thermal performance of the phenolic thermal protection material?
- How do the curing conditions (temperature, pressure, heating rate) affect the long-term durability and performance of the final phenolic composite in extreme aerospace environments?
- How do variations in resin flow and fiber compaction influence the formation of micro-defects, and what strategies could be implemented to further reduce porosity?
- Could the curing process model developed for filament-wound preforms be adapted for other composite manufacturing techniques, and what modifications would be needed?
Author Response
We sincerely appreciate the reviewer’s constructive comments and insightful questions regarding our manuscript titled “A Study on the Compaction Behavior and Parameter Sensitivity of Curing Phenolic Thermal Protection Material Strips.” We have carefully considered each point and revised the manuscript accordingly. Below, we provide detailed responses to the reviewer’s questions, with references to the relevant sections and figures in the revised version.
Comment 1
How do the observed variations in resin viscosity during the curing process influence the final mechanical properties and thermal performance of the phenolic thermal protection material?
Response:
The viscosity evolution of phenolic resin critically affects fiber impregnation quality, void formation, and internal stress distribution, which are key determinants of the composite’s mechanical and thermal properties. Our study shows that the resin exhibits a typical U-shaped viscosity–time profile during curing. If the viscosity remains high during early heating, incomplete fiber wetting and dry regions form, weakening the interfacial bonding. Conversely, rapid heating may cause premature gelation, trapping volatiles and leading to voids.
As described in Section 3.1, the DSC results demonstrate thermal lag under fast heating conditions, along with a reduction in total reaction heat (ΔH) and a narrower reaction window (ΔT), indicating incomplete curing. This underlines the importance of controlling the low-viscosity window to optimize flow, compaction, and volatile removal. Voids formed during curing, observed in Section 3.3, indirectly indicate compromised mechanical and thermal performance, consistent with previous studies.
Comment 2
How do the curing conditions (temperature, pressure, heating rate) affect the long-term durability and performance of the final phenolic composite in extreme aerospace environments?
Response:
Curing conditions directly influence crosslink density, porosity, and fiber–matrix interfacial quality, which govern long-term thermal and mechanical stability in aerospace applications. Adequate curing temperature ensures full crosslinking and improved oxidation resistance. However, excessively fast heating or high temperatures can result in internal thermal gradients and premature surface curing, which hinders gas escape and promotes void formation (Section 3.3, Figure 12).
Curing pressure improves resin flow and interlaminar compaction. As shown in Figure 14, pressure increases from 0.4 MPa to 0.6 MPa significantly reduce void content, leading to a denser, more reliable laminate. Section 3.2 further confirms that the fiber volume fraction—a key performance indicator—reaches optimal values under curing temperatures of 165–180 °C, heating rates of 2.5–4.5 °C/min, and pressures of 0.4–0.6 MPa. These conditions are proposed as a robust process window for achieving long-term durability.
Comment 3
How do variations in resin flow and fiber compaction influence the formation of micro-defects, and what strategies could be implemented to further reduce porosity?
Response:
Resin flow and compaction behavior strongly affect void nucleation. Insufficient resin mobility or uneven compaction leads to fiber dry zones and gas entrapment, forming ellipsoidal voids as observed in Section 3.3 (Figures 12–14). At low curing temperatures, prolonged low viscosity allows better gas escape, whereas higher temperatures may cause superficial gelation that traps volatiles in the laminate core. These trends are consistent with the findings of Pupin et al. [19] and Mehdikhani et al. [22].
To further reduce porosity, we propose the following strategies:
(a) Reduce heating rates to extend the degassing period before gelation;
(b) Dynamically apply curing pressure synchronized with the viscosity drop;
(c) Incorporate breather and bleeder layers to enhance venting efficiency.
These suggestions are discussed in Section 3.3 and supported by the void evolution mechanism illustrated in Figure 15.
Comment 4
Could the curing process model developed for filament-wound preforms be adapted for other composite manufacturing techniques, and what modifications would be needed?
Response:
Yes, the curing process model developed in this study, which couples heat conduction, viscosity evolution, and resin flow–compaction behavior, can be extended to other thermoset composite manufacturing methods, such as autoclave molding, compression molding, and VARTM.
However, the model must be adjusted for different boundary conditions. For example, in compression molding, uniform pressure requires reformulation of the compaction equation, while in VARTM, resin infusion and capillary flow must be included using Darcy’s law.
Additionally, as noted in Section 3.5, our experiments focus on flat laminates. Real-world aerospace structures often involve complex geometries and thickness variations, requiring further validation. Future work will focus on applying the model to curved components, integrating multi-field simulations, and linking macroscopic process models with microstructural evolution and performance predictions.
We again thank the reviewer for the thoughtful suggestions, which have helped us to significantly improve the clarity and scientific rigor of the manuscript.
Sincerely,
Zeyu Pan
(On behalf of all authors)
Reviewer 2 Report
Comments and Suggestions for Authors
Review of the manuskrtype - ID: polymers-3547983
A Study on the Compaction Behavior and Parameter Sensitivity of Curing Phenolic Thermal Protection Material Strips
The article submitted for review deals with the curing process of filament-wound phenolic resin. The Authors have properly conducted a multidirectional study. The methodology was presented in a clear manner, with important points made to allow repetition of the study by other researchers, which is crucial for the credibility and transparency of the science. The introduction was also well edited and the purpose and scope were precisely outlined. Unfortunately, the rest of the article deviates significantly from the standards of scientific publications and needs significant improvement.
- The manuscript contains numerous editorial errors that detract from its professional nature. Note the inconsistencies in text formatting and diagrams. These errors make it difficult to read and the reception of the work as a whole is unprofessional.
- The analysis of the results is more like a research report than a scientific article. The results section lacks reference to current knowledge and scientific literature, which indicates insufficient discussion of the results. The Authors limited references to the literature to the Introduction only, which is unacceptable in a scientific paper, where it is important to place one's own results in the context of previous research. This is all the more important since the research was conducted in an area of already well-known knowledge and wide access to the literature.
- In the context of the above, the analysis of the results completely ignores the discussion of their significance in relation to existing knowledge. The Authors did not attempt to compare their results with those of other researchers, nor did they undertake a critical analysis of their own results, which makes the article incomplete and of little scientific value. Moreover, some issues studied by the Authors - such as DSC analysis of the resin under different temperature conditions - are already well known in the literature. I think it is necessary for the Authors to take into account the existing studies in this field and point out the original contribution of their work. This remark also applies to the analysis of the effect of curing process parameters on pore distribution
- In the analysis of the results, many times the Authors treated the description of the observed phenomenon in a superficial way, without a deeper analysis and understanding of the mechanisms responsible for this phenomenon. An example is the sentence on page 12, lines 310-316. What scientific value does this sentence have?
Conclusions:
Although the manuscript was based on correctly conducted research, its final form does not meet the standards of scientific publication. For the work to be published, the following changes are necessary:
- Correcting editorial errors and ensuring professional editing of the text.
- Expanding the analysis of the results by placing them in the context of the scientific literature, conducting a fair discussion and indicating the original contribution of the work in relation to already known data.
- Extend references to the scientific literature to other sections of the article, rather than limiting them to the introduction only.
- Pointing out the original elements of the work, especially with regard to issues that are already well known in the literature.
- Detail in scientific detail the explanation of critical phenomena observed in Author’s study, such as pore formation during resin curing.
Recommendations - a major revision

needs a major revision
Author Response
Dear Reviewer,
We sincerely appreciate your time and effort in reviewing our manuscript entitled:
“A Study on the Compaction Behavior and Parameter Sensitivity of Curing Phenolic Thermal Protection Material Strips”.
Thank you for your constructive and insightful comments, which have helped us significantly improve the quality of the manuscript. In response to your suggestions, we have carefully revised the manuscript and addressed each concern point-by-point as follows. All modifications have been marked in yellow in the revised manuscript.
Comment 1:
“The manuscript contains numerous editorial errors that detract from its professional nature. Note the inconsistencies in text formatting and diagrams.”
Response:
Thank you for your valuable comment. We have thoroughly revised the manuscript to correct all formatting inconsistencies. Specific actions include:
- Standardizing figure references (using “Figure” uniformly);
- Re-numbering all figures and tables to ensure logical order;
- Unifying fonts, paragraph spacing, and styles throughout the manuscript;
- Aligning the layout strictly with the Polymers journal template.
These improvements greatly enhance the manuscript’s readability and professional presentation.
Comment 2:
“The analysis of the results lacks reference to current knowledge and scientific literature. The authors limited references to the literature to the Introduction only, which is unacceptable in a scientific paper.”
Response:
We fully agree with your observation. In the revised manuscript, we have significantly expanded the “Results and Discussion” section by integrating relevant and up-to-date scientific literature throughout, rather than limiting it to the Introduction.
Key modifications include:
- In Section 3.1 (DSC Analysis), we added a detailed discussion supported by the works of Dusi et al. [8], Guo et al. [15], and Park et al. [25], to explain the thermal lag behavior under different heating rates.
- In Section 3.2, our findings on fiber volume fraction changes were linked to the compaction theory of laminated composites proposed by Gutowski et al. [14,21].
- In Section 3.3, the analysis of void evolution was discussed in relation to the work of Pupin et al. [19] and Mehdikhani et al. [22], emphasizing the effects of premature surface curing and internal thermal gradients.
These revisions ensure that our findings are placed in the context of existing research, strengthening the scientific grounding of the manuscript.
Comment 3:
“The analysis ignores the significance of results in relation to existing knowledge. No critical comparison or evaluation is presented, making the article incomplete and of limited scientific value.”
Response:
Thank you for this important comment. In response, we made the following major improvements:
- Scientific Comparison with Prior Work:
Across Sections 3.1 to 3.4, we now systematically compare our experimental results with existing knowledge in curing kinetics, void formation mechanisms, resin flow theory, and compaction behavior. This establishes the novelty and significance of our findings. - Critical Reflection and Limitations:
We have added a new section, 3.5 Limitations and Future Perspectives, which critically discusses the following aspects: - Applicability of the model to complex geometries;
- Limitations of the empirical regression model;
- Limitations of 2D void analysis methods and the potential of 3D μCT techniques;
- Influence of unconsidered variables (e.g., humidity, mold properties, degassing);
- Lack of real-time monitoring and suggested future directions (e.g., dielectric analysis, FBG sensing).
This section provides thoughtful evaluation and outlines practical and academic directions for future research, enhancing the completeness and scientific value of the work.
Comment 4:
“Some results are described superficially, without deeper analysis or understanding of the underlying mechanisms. For example, the sentence on page 12, lines 310–316, lacks scientific value.”
Response:
We appreciate your suggestion and have made the following specific revisions:
- Rewritten superficial descriptions:
The sentence on page 12 has been completely revised based on a mechanism-driven explanation involving temperature gradients, resin gelation timing, and gas entrapment. Literature by Agius et al. [21] was cited to support the observed mechanism of surface layer encapsulation causing void formation. The updated version is now located on page 13, lines 339–347. - Improved similar sections (e.g., page 14, lines 361–373):
We analyzed marginal pressure effects on consolidation and discussed the void gradient distribution across thickness. These insights are supported by the findings of Gutowski et al. [14] regarding pressure-induced void localization. - Incorporated void nucleation theory:
At the end of Section 3.3 (page 14, lines 367–383), we introduced classic theories of void nucleation in thermosetting composites and discussed how thermal gradients, local pressure fields, and resin viscosity influence void formation and morphology, supported by Pupin et al. [19]. - Global improvements in scientific rigor:
We reviewed the entire manuscript to ensure that every experimental observation is interpreted through a mechanistic or literature-backed framework, eliminating any vague or unsupported assertions.
Conclusion:
We once again thank the reviewer for the thoughtful and detailed feedback. Your suggestions have led to a significantly improved version of our manuscript, both in scientific depth and presentation quality. We hope that the revised manuscript meets your expectations and is now suitable for publication.
Sincerely,
Zeyu Pan
On behalf of all authors

Round 2
Reviewer 1 Report
Comments and Suggestions for Authors
Authors answerred alll my questions. I think the manuscript is ready for publication.
Author Response
We sincerely thank the reviewer for their positive comments and recommendation. We appreciate your time and effort in reviewing our manuscript. Thank you again for your constructive feedback and support throughout the review process.
Reviewer 2 Report
Comments and Suggestions for Authors
Re-review of manuscript ID: polymers-3547983
The Authors have made substantive corrections to the manuscript, which should be appreciated. However, despite earlier attention to serious editorial shortcomings, corrections in this regard were not carried out. The failure to make these changes indicates insufficient adherence to editorial standards, which negatively affects the professionalism of the article. Therefore, the text still requires thorough editorial corrections to meet the requirements of a scientific publication.
Comments on the Quality of English Language
correction by a native speaker is required
Author Response
Thank you for your continued feedback and for recognizing the substantive improvements in our revised manuscript. We sincerely apologize for not fully meeting the editorial and language standards in the previous version. In response to your comment, we have now carefully revised the entire manuscript for language clarity, grammar, and expression.
To ensure the highest linguistic quality, the revised manuscript has been thoroughly proofread by a professional native English speaker with experience in scientific writing. All grammatical inconsistencies, awkward phrasings, and stylistic issues have been addressed. We believe the current version of the manuscript now meets the standards of academic English required for publication.
Thank you for your patience and valuable guidance.
Round 3
Reviewer 2 Report
Comments and Suggestions for Authors
Despite twice drawing attention to the poor editing of the manuscript, the authors still have not thoroughly revised the text. This leads to numerous editing shortcomings, such as inconsistent font, formatting errors, and inconsistent citation style. In its current form, the manuscript gives the impression of being underdeveloped and requires careful technical editing to meet publication standards.
The lack of response to previous comments suggests insufficient commitment by the authors to improve the quality of the text. Again, I encourage a thorough revision and a more careful approach to editing the manuscript.
Comments on the Quality of English Language
Text still needs language correction.